# Line-YOLO: An Efficient Detection Algorithm for Power Line Angle

**DOI:** 10.3390/s25030876

**Published:** 2025-01-31

**Authors:** Chuanjiang Wang, Yuqing Chen, Zecong Wu, Baoqi Liu, Hao Tian, Dongxiao Jiang, Xiujuan Sun

**Affiliations:** 1College of Electrical Engineering and Automation, Shandong University of Science and Technology, Qingdao 266590, China; cxjwang@sdust.edu.cn (C.W.); cyq010801@163.com (Y.C.); liu687666@163.com (B.L.); th989898thth@163.com (H.T.); 2Da Tuo (Shandong) Internet of Things Technology Co., Ltd., Qingdao 266555, China; wu17860763313@163.com; 3Engineering Technology Company, Shandong Airlines Co., Ltd., Qingdao 266300, China; jiangdxsda@163.com

**Keywords:** power line detection, YOLOv8s-seg algorithm, deformable convolution, BiFPN feature fusion, EMA attention mechanism

## Abstract

Aiming at the problem that the workload of human judgment of the power line tilt angle is large and prone to large errors, this paper proposes an improved algorithm Line-YOLO based on YOLOv8s-seg. Firstly, the problem of the variable shape of the power line is solved through the introduction of deformable convolutional DCNv4, and the detection accuracy is improved. The BiFPN structure is also introduced for the Neck layer, which shortens the time required for feature fusion and improves the detection efficiency. After that, the EMA attention mechanism module is added behind the second and third C2f modules of the original model, which improves the model’s ability to recognize the target, and effectively solves the problem of loss and error when power line targets overlap. Finally, a small target detection head is added after the first EMA attention mechanism module for detecting small or occluded targets in the image, which improves the model’s ability to detect small targets. In this paper, we conduct experiments by collecting relevant power line connection images and making our dataset. The experimental results show that the mAP@0.5 of Line-YOLO is improved by 6.2% compared to the benchmark model, the number of parameters is reduced by 28.2%, the floating-point operations per second is enhanced by 35.3%, and the number of detected frames per second is improved by 14 FPS. It is proved by the experiments that the enhanced model Line-YOLO detects the results better, and it can efficiently complete the power line angle detection task.

## 1. Introduction

Distribution cabinets are widely used in critical facilities to ensure power supply and distribution [1]. In these cabinets, the quality of electrical connections directly affects the stability and safety of power transmission, which in turn influences the power transfer efficiency, load balance, and fault frequency. This highlights the importance of connection quality. When evaluating electrical connections, the angle of the power line is a key parameter. Both excessively large and small angles can lead to issues such as load imbalance, voltage loss, and power line overload. Power line angle detection refers to accurately measuring the tilt angle of power lines in space using various technologies. The core objective is to control the angle difference between the power line and a reference line within a reasonable range, ensuring stable system operation. This provides foundational data for subsequent power line monitoring, maintenance, and automated inspections. Historically, the manual measurement of power line angles was prone to significant subjective errors, leading to inconsistencies and inaccuracies due to varying results from different operators. Therefore, new methods and technologies are needed to achieve accurate and reliable measurements of power line connection angles.

Deep learning is a data-driven approach. To train a network model capable of predicting the type and location of the target, a large amount of image data must be input into the network. This enables the computer to automatically learn the features of the images and improve the results by reducing the gap between predicted and actual values [2]. Deep learning algorithms can effectively address the low efficiency of manual detection. Some researchers have begun exploring the use of deep learning for power line detection tasks. While these methods and technologies differ, their common goal is to improve the accuracy and efficiency of power line segmentation. Algorithms such as the two-stage Faster R-CNN [3] and one-stage algorithms like SSD [4] and YOLO exhibit excellent detection performance. Integrating deep learning into detection workflows reduces labor costs, enhances efficiency, and minimizes errors. Therefore, applying deep learning for power line angle detection is of significant research importance.

Integrating object detection algorithms with power system monitoring can enable the real-time monitoring of power line angles, allowing for the timely identification and resolution of potential issues. By calculating the power line connection angles, the operational status of the power lines can be monitored and analyzed in real-time. This provides data support for grid planning and upgrades, enhancing the overall efficiency and safety of the power system. This approach not only improves the efficiency of power line connections but also provides strong technical support for the maintenance and optimization of the power system.

The main focus of this study is the application of deep learning algorithms to accurately segment power lines in target images. At the same time, the aim is to reduce the computational cost of the model, enabling more efficient use of computational and memory resources without sacrificing accuracy. The purpose of this research is to explore how to more effectively apply deep learning models to distinguish power lines from their background. The goal is to enhance the segmentation accuracy and robustness of power lines while reducing the model size. We propose an improved model, Line-YOLO, based on YOLOv8s-seg. This model incorporates deformable convolutions, the BiFPN feature pyramid network architecture, EMA attention mechanisms, and a small object detection head to achieve the precise segmentation and accurate angle measurement of power lines in target images. The main research topics include the following:(1)Self-made dataset: Due to the lack of publicly available relevant datasets, this study creates a custom dataset using images captured from electrical worker examination objects provided by a power grid company, as well as images collected from the internet. The quality of the dataset directly impacts the model’s performance. To enhance dataset quality, data augmentation techniques are applied to the images, and pixel-level annotations and classifications are performed. This provides strong data support for subsequent model training and testing.(2)Improvement of the baseline model: Several modules in the YOLOv8s-seg baseline network model are optimized. These improvements include the C2f_DCNv4 module, BiFPN structure, EMA attention mechanism, and a small object detection head. These modifications enhance the model’s generalization ability, training speed, and detection performance for small objects.(3)Experimental results and angle measurement: Single-module comparison experiments are conducted to evaluate each modification to the baseline model. Additionally, comparative and ablation experiments are performed on the improved model. Finally, based on the detected power line positions, the horizontal and vertical angles of the power lines are measured.

## 2. Related Works

There is limited research on power line angle detection, but numerous studies focus on power line detection. Power line angle detection is a more advanced task based on power line detection. It depends on the results of power line detection. Only by first identifying the exact locations of power lines can angle measurements be performed at those positions. Therefore, accurate power line detection is a prerequisite for power line angle detection.

### 2.1. Power Line Detection

In traditional digital image processing, researchers often utilize features such as image edge characteristics, the linear shape of power lines, and the contrast between power lines and their surrounding environment to filter and identify target power lines. Liu et al. [5] employed edge detection through edge map drawing techniques and applied an improved Hough transform for region partitioning to effectively extract power lines. However, this method performs poorly when power lines bend due to factors such as gravity. Song et al. [6] calculated the first derivatives of the matched filter and Gaussian filter, then combined them for edge detection. The detected results were then input into a morphological filter to select power lines. However, this approach does not effectively distinguish power lines from other background information. Chen et al. [7] identified power lines by comparing the similarity of backgrounds on both sides of the line structure. However, due to the complex backgrounds around power lines, this method may lead to misidentification. Jiao et al. [8] attempted to locate power lines by combining the directionality and spectral features of power lines with the Hough transform, but this strategy is unstable in non-specific environments. Xiong et al. [9] proposed a new method for millimeter-wave radar-based power line detection and classification using the Hough transform and convolutional neural networks, but it demonstrated poor detection accuracy.

In recent years, with the continuous development of deep learning, researchers have begun applying deep learning to power line detection. For example, Yetgin et al. [10] developed a model using convolutional neural networks to detect the presence of power lines in images, but it cannot accurately locate them. Zhang et al. [11] used a cropped VGG16 network as the base, supplemented with a decoder, but the limited size of their dataset restricted its application in diverse power transmission landscapes. Choi et al. [12] proposed a weakly supervised learning strategy to reduce the labeling effort required for dataset creation, thus lowering the cost of detection algorithms. Long et al. [13] employed a fully convolutional network (FCN) for power line segmentation, but the segmentation performance still requires improvement. Some researchers used physical rendering techniques in synthetic image creation, developing an innovative LS-Net single-stage detection framework for efficient power line detection. However, during training, the embedded classification sub-network and line segment regression component lacked sufficient adaptability [14]. The design of the DeepLab v3+ [15] decoder enables more precise image segmentation, but it results in a significant increase in model parameters. Zhu et al. [16] developed a fast power line detection network called Fast-PLDN, which effectively reduced errors in power line boundary detection. However, its performance decreased when power lines were too close to each other. Overall, deep learning-based methods outperform traditional digital image processing techniques in terms of both accuracy and robustness for power line image segmentation, making deep learning widely researched.

### 2.2. Algorithm Development

To ensure efficient power line detection, faster detection algorithms, such as one-stage algorithms, should be selected. Representative algorithms include SSD, RetinaNet [17], and the YOLO series. The SSD algorithm, introduced in 2016, offers a balanced trade-off between detection accuracy and speed. However, it uses the standard intersection over union (IoU) for object matching, which may lead to an imbalanced positive-to-negative sample ratio, affecting training performance. The RetinaNet algorithm introduces focal loss to address the class imbalance problem, achieving high detection accuracy but with greater computational demands and slower detection speeds. In 2015, Joseph Redmon and his team introduced YOLOv1 [18], a real-time object detection algorithm based on convolutional neural networks. YOLOv2 [19], the second version in the YOLO series, introduced several innovations, such as deeper convolutional network architectures, the use of Anchor Boxes to replace earlier grid prediction methods, and the inclusion of batch normalization to reduce overfitting risks. YOLOv3, which followed, outperformed YOLOv2 [20] in key aspects such as detection accuracy, multi-scale detection capability, robustness, and processing speed.

To improve model efficiency, network structures are often optimized. FFCA-YOLO [21] reconstructs the backbone and Neck layers using partial convolutions, reducing frequent memory redundancy accesses. GCL-YOLO [22] constructs a backbone based on GhostConv, generating redundant feature maps to minimize detection accuracy loss. MELF-YOLOv5 [23] uses MobileNetV3 [24] as the backbone and employs depthwise separable convolutions to control the number of network channels, reducing the model size and computational load. PP-PicoDet [25] reduces parameters by unifying the number of input channels and extends the use of larger separable convolutions to improve accuracy. INAR-SSD [26] introduces the GoogLeNet Inception structure and Rainbow concatenation to propose a deep CNN-based detection model with high detection accuracy and speed. Faster-RCNN-SE-FA [27] applies an SE module to enhance the distinction of features in the channel dimension while retaining position sensitivity, improving detection performance. Liang et al. [28] combined the parallel auxiliary multi-scale feature enhancement module (MFEM) with RetinaNet to construct a model with a bidirectional feature pyramid network. This model effectively integrates strong semantic information from high-level networks and high-resolution information from low-level networks, improving detection performance for small objects. Sithmini et al. [29] introduced the Xception architecture, combining depthwise separable convolutions and point convolutions to enhance feature representation learning. This method achieves better performance at the same computational complexity.

Based on comparisons of various algorithms, this study selects YOLO series algorithms due to their high detection accuracy and speed, stable updates, wide applicability, and suitability for practical use.

## 3. Proposed Approach

### 3.1. Proposal Network Overview

YOLOv8 was introduced by Glenn Jocher, the founder of Ultralytics and the author of YOLOv5. It significantly improves object detection performance by adopting a more efficient network architecture. The YOLOv8 model has five different architectures: n, s, m, l, and x. The primary differences between these architectures lie in the model depth of the submodules and the number of channels in the convolutional layers. After comparing the depth of the model and the width of the feature, YOLOv8s is more suitable for power line detection tasks. YOLOv8s-seg is the segmentation version of YOLOv8s, focusing on both object detection and segmentation tasks. This makes it particularly well suited for power line detection, as YOLOv8s-seg can not only detect the location of objects but also perform pixel-level segmentation. This enables more precise identification of the power line’s specific area, which is crucial for applications that require fine distinction between power lines and the background. Additionally, compared to standard object detection tasks, the YOLOv8s-seg model can identify the exact contours and shapes of objects. In power line detection, especially in complex backgrounds, precise boundary detection helps better locate power lines and reduces the probability of false positives and missed detections.

This study improves the baseline YOLOv8s-seg network by adding the deformable convolution DCNv4 to the second and third C2f modules in the feature extraction network, forming the C2f_DCNv4 module. This enhancement allows the model to capture image features more finely, supporting feature fusion and subsequent predictions in the detection head. Additionally, the improved BiFPN structure is fused with the Concat structure to create a new BiConcat module, replacing all Concat modules. This facilitates the exchange and fusion of high-level and low-level feature information, improving detection accuracy. The study also adds an EMA attention module after the third and fourth C2f modules in the feature fusion layer. Furthermore, a small target detection head of 160 × 160 is added after the first EMA attention module. This modification increases the use of contextual information, improving the model’s ability to detect small targets and effectively addressing issues related to target overlap, such as loss and misjudgment. By incorporating these improvement modules into the baseline model, we develop a new power line detection network framework, referred to as the Line-YOLO model, which is used for experimental research in power line detection. Its network structure is shown in Figure 1.

### 3.2. Deformable Convolution

In order to solve the problem of complex and diverse power lines in distribution cabinets, this paper incorporates deformable convolution into the algorithm. The use of deformable convolution instead of traditional convolution can achieve the enhancement of its ability to recognize geometric transformations so that the model can more accurately identify the deformation of the object [30]. Moreover, the deformable convolution module can easily replace the common module in the existing CNN [31]. The deformable convolution output feature values are shown in Equation (Equation 1), where *G* denotes the number of group aggregations, and for the g group, $w_{g}\in R^{C\times C^{\prime }}$ denotes the position-independent mapping weight of the group, where $C^{\prime }=\frac{C}{G}$ denotes the group dimension. $m_{\mathrm{gk}}\in R$ denotes the modulation scalar of the *k* sampling point of the g group, normalized by the Softmax function along the dimension *K*. $x_{g}\in R^{C^{\prime }\times H\times W}$ denotes the input feature map of the slice, and $\Delta p_{\mathrm{gk}}$ is the g group grid offset corresponding to the sampling position $p_{k}$:(1)$$y\left(p_{0}\right)=\sum_{g=1}^{G}\sum_{k=1}^{K}w_{g}\cdot m_{\mathrm{gk}}\cdot x_{g}\left(p_{0}+p_{k}+\Delta p_{\mathrm{gk}}\right)$$

The deformable convolution DCNv4 used in this study improves upon the previous version, DCNv3. First, it removes the Softmax normalization in spatial aggregation, enhancing its dynamic properties and expressive capabilities. Second, it optimizes the memory access process, accelerating the processing speed. Compared to DCNv3, DCNv4 uses a single thread to process multiple channels within the same group, with these channels sharing sampling offsets and aggregation weights. This method reduces the workload of memory reading and bilinear interpolation coefficient calculations. It also allows for the merging of multiple memory access instructions, enabling DCNv4 to reduce memory access output tensors by half. The optimization not only improves processing speed but also conserves computational resources. In this study, DCNv4 is integrated into the YOLOv8 network by merging the second and third C2f modules of the backbone network with the deformable convolution module, particularly focusing on the convolutional layers of P3 and P4 detection levels. This strengthens the model’s ability to recognize small- and medium-sized targets. The improved C2f module is named C2f_DCNv4. After introducing the C2f_DCNv4 module, the receptive field for small target locations can be adaptively adjusted, allowing the model to more accurately fine-tune its regression parameters during the prediction box regression. This results in better focus on small targets and ultimately improves the overall performance of the model. The structure of the C2f_DCNv4 module is shown in Figure 2.

### 3.3. Feature Fusion

The feature fusion of YOLOv8 adopts the FPN+PAN structure. Because of the diversity of power line target sizes, in this paper, the Neck layer is optimized, and the BiFPN structure is adopted to improve the detection performance [32]. BiFPN employs a novel approach in the feature fusion stage, which assigns an additional weight to each of the input feature maps at different scales, adjusting the contribution of the feature maps at each scale, and realizing the weighted fusion [33]. The structure utilizes a fast normalized fusion method that aims to reduce the time required for feature fusion. As shown in Equation (Equation 2), $w_{i}$ represents the weights after processing by the ReLU function:(2)$$O=\sum_{i}\frac{w_{i}}{\varepsilon +\sum_{j}w_{j}}I_{i}$$

In the pursuit of efficiency improvement, BiFPN identifies that the impact of individual input nodes on the fusion effect is negligible. Therefore, nodes with a single input are removed. Simultaneously, BiFPN introduces direct connections between the input and output layers, and uses inter-layer skip connections to promote the flow of cross-channel information. Although this slightly increases computational cost, it enables deeper integration of features, thereby enhancing the model’s ability to process features. Retaining too many shallow feature details might result in the loss of richer deep semantic features, so it is beneficial to assign higher weights to deeper features. Since the YOLO model primarily utilizes the P3 to P5 layers for feature fusion, and smaller targets have lower visibility—rarely raising concerns about privacy leakage—this study simplifies the application of BiFPN by restricting the weighted feature fusion to only the P3 to P5 layers. Additionally, to improve both efficiency and effectiveness, a "shortcut" connection is retained at the P4 layer. The structures of the BiFPN before and after the improvement are shown in Figure 3, where (a) represents the BiFPN structure before the improvement and (b) represents the structure after the improvement.

### 3.4. Attention Mechanism

When detecting power lines in a distribution cabinet, the image background is complex, and there is a significant amount of interference. To allow the network model to focus on the location of the power lines during detection, this paper introduces an attention mechanism to address this issue. Among various attention mechanisms, the STN attention mechanism [34] adapts the spatial layout by learning the geometric transformation parameters of the input image. However, it shows limitations in capturing long-range dependencies. The CA attention mechanism [35] primarily focuses on the relationship between pixels within the image features. By examining the relationship between the position information and feature values in the feature map, it can identify long-distance spatial interactions. However, this method does not fully consider the importance of interactions between spatial positions. Jie et al. [36] proposed SENet, which uses a channel attention mechanism that significantly models the dependencies between different feature channels. It autonomously identifies the importance of each channel through the learning process. However, it mainly focuses on the relationships between feature channels and neglects the information in the spatial dimension.

In this paper, we choose to use the EMA attention mechanism [37]. As a technique based on cross-space learning, EMA employs an efficient multi-scale attention architecture that can be implemented through a grouping structure without requiring dimensionality reduction. It deals with the two dimensions of height and width of the feature map and applies two-dimensional average pooling to improve the computational efficiency. The pooling operation is shown in Equation (Equation 3).Where *H* and *W* refer to the height and width of the feature map respectively, while $x_{c}$ represents the feature tensor on different channels:(3)$$Z_{c}=\frac{1}{H\times W}\sum_{j}^{H}\sum_{i}^{W}x_{c}\left(i,j\right)$$

Through cross-spatial learning, the EMA attention mechanism expands the feature space, efficiently capturing the interdependencies between the three channels while reducing computational costs and preserving spatial structure information. The output features of the three channels are a combination of two spatial attention weight values, which, after activation by the Sigmoid function, enhance the relationships between contextual pixels. Additionally, by using 3 × 3 and 1 × 1 convolution kernels in parallel to process medium-sized feature maps, the use of contextual information is increased. This not only improves the model’s ability to recognize objects but also effectively addresses the issues of loss and misjudgment during target overlap. This paper opts to add the EMA attention module after the third and fourth C2f modules in the feature fusion layer. The structure of the EMA attention mechanism is shown in Figure 4.

### 3.5. Target Detection Head

The main difficulty in power line detection tasks lies in the stacking of line clusters and severe occlusion between power lines, which can result in the loss of feature information. Zhu et al. [38] introduced a Transformer-based detection head in their model, which showed excellent performance in handling densely packed objects with significant scale variation under high-speed flight or low-altitude shooting conditions. However, this approach increases the computational demand of the model. The baseline model YOLOv8s-seg can output feature maps of three sizes: 20 × 20, 40 × 40, and 80 × 80. However, under this configuration, detecting smaller power line targets remains challenging.

In this paper, a new detection head of size 160 × 160 is added to the basic structure of the YOLOv8s-seg model to improve detection accuracy for smaller targets. The YOLOv8s-seg model extracts initial features at the sixth layer of the backbone, and the shallow features are fused with contextual information extracted via the EMA attention mechanism in the Neck layer using the Concat operation. The newly added detection head is located at the output of the first EMA attention module, specifically for small target detection. Although this improvement slightly increases the model’s computational load, it significantly enhances the model’s ability to detect small-sized targets and effectively reduces false positives and false negatives at different scales. The improved detection layer structure is shown in Figure 5.

## 4. Experiment

### 4.1. Dataset Description

The experiments in this paper are conducted using a self-made dataset. The original images include both power line images provided by the power grid company for electrician exams and relevant images collected from the internet. Data augmentation techniques such as flipping, rotating, scaling, cropping, adjusting brightness, and image aliasing are applied to these images, resulting in a total of 2700 images. The images are annotated uniformly according to their serial numbers. These images are then divided into training, validation, and test sets with an 8:1:1 ratio to ensure sufficient training data.

Data annotation is performed using the Labelme tool, with pixel-level annotations applied to all obtained image features. The images are cropped to a consistent size of 640 × 640 pixels, and those with severe occlusion or disordered power lines are discarded. This cleaning process ensures the remaining dataset is more suitable for model training and prediction.

### 4.2. Implementation Details and Evaluation Indicators

For the experiments, this paper uses Python 3.9, PyTorch 2.1.0, and CUDA 12.1 as the software environment, Tesla V100S-PCIE-32GB for the graphics card, Intel(R) Gold 6226R CPU @ 2.90 GHz for the processor, and Windows 11 for the operating system. The experiments are required to maintain the consistency of all the parameters in order to ensure fairness; pre-trained weights are not used in all experiments. The specific hyperparameter settings for the experiment are shown in Table 1.

When performing target detection, the accuracy of operations such as classification, localization, and operation are usually used to evaluate the merits of the network model. For the power line dataset, the commonly used evaluation metrics in this paper are as follows: precision, recall, AP (Average Precision), mAP (mean Average Precision), parameter, and FLOPs (floating-point operations per second).

Precision (*P*) is a metric for assessing the accuracy of the model, and recall (*R*) represents the proportion of correctly predicted positive samples to the total number of positive samples, which are calculated as shown in Equations (Equation 4) and (Equation 5). Among them, TP, FP, and FN stand for true examples, false positive examples, and false negative examples, respectively. We have the following:(4)$$P=\frac{TP}{TP+FP}$$(5)$$R=\frac{TP}{TP+FN}$$

AP can be used as a modeling measure and is more applicable in single-class testing. In the coordinate system, *R* is the horizontal coordinate and *P* is the vertical coordinate. The PR curve is derived from its value, and the area enclosed by the curve and the horizontal coordinate is the AP metric, which is calculated as shown in Equation (Equation 6):(6)$$AP=\int_{0}^{1}P\left(R\right)dr$$

The mAP further averages the mean values after the detection of each target category, which can evaluate the performance of the whole target detection system. After obtaining the average accuracy of a single category, the mean Average Precision can then be obtained by the sum and average operation, which is calculated as shown in Equation (Equation 7). Here, $AP_{i}$ denotes the average accuracy of the i-th category, and *n* denotes the number of categories targeted for detection:(7)$$mAP=\frac{1}{n}\sum_{i=1}^{n}AP_{i}$$

### 4.3. Comparative Experiments with Improved Modules

#### 4.3.1. Effects of the C2f_DCNv4 Module

After introducing the improved C2f_DCNv4 module into YOLOv8s-seg, feature extraction and model performance are significantly enhanced, improving the detection accuracy and robustness. The improved module also strengthens the detection of small targets, allowing the model to more accurately locate and identify small-sized objects, achieving ideal detection results even in complex backgrounds. In this study, the detection results of the YOLOv8s-seg model with the introduced C2f_DCNv4 module are compared with those of YOLOv5s-seg, YOLOv7s-seg, and the baseline YOLOv8s-seg model. The experimental results are shown in Table 2. As seen in Table 2, after incorporating the C2f_DCNv4 module, the model’s mAP@0.5 is increased by 2.1%. Although the computational complexity is slightly increased, the frames processed per second are improved, leading to a significant enhancement in the model’s computational efficiency and inference speed.

#### 4.3.2. Effects of the BiFPN Module

Different fusion strategies for multi-scale fusion can have varying impacts on the model’s performance. In the weight fusion strategy, each feature layer undergoes convolution to enhance its representational capability. The layers are then processed using normalization methods and the RELU function to obtain a weight vector. These weights are normalized and applied through multiplication, and the weighted feature layers are stacked along specific dimensions. The stacked feature layers are then summed to produce the final output feature map. In the Sum fusion strategy, the convolved feature layers are stacked along specific dimensions, and the resulting feature map is generated and used as the final output. In the adaptive fusion strategy, the feature layers are concatenated along the channel dimension. The Softmax function is then used to generate adaptive weights for the concatenated feature layers, which are processed following the same steps as the Weight strategy. The Concat fusion strategy involves a simple concatenation process, where all convolved feature layers are directly concatenated along the channel dimension, and the output is produced after fusion of the layers. By adding the BiFPN module to the YOLOv8s-seg baseline model and testing with the four fusion strategies, the results are shown in Table 3. As seen in Table 3, when the BiFPN module is combined with the Concat module, the model achieves the highest mAP@0.5 of 87.1%, with the fewest number of parameters. Therefore, the introduction of the fused BiConcat module effectively improves the model’s accuracy and detection speed.

#### 4.3.3. Effects of the EMA Module

When evaluating the improvement effects of various attention mechanisms, although SENet contributes only slightly to enhancing the model’s accuracy (an increase of 0.2%), it significantly increases the model’s parameter count. ECANet, on the other hand, improves precision (an increase of 1.6%), but this also leads to an increase in the model’s computational demands. Despite the widespread use of the CBAM attention mechanism in many studies, it is not suitable for handling the complex power line connections in this case and thus does not improve the model’s accuracy. The Line-YOLO algorithm adopts an efficient multi-scale attention mechanism across space and explores two improvement paths in this study. The first approach involves combining the EMA module with the C2f module from YOLOv8s-seg to form the C2f_EMA module. However, this improvement does not show a significant boost in accuracy and does not fully exploit its potential. The second approach involves directly feeding the output information from the C2f module into the EMA module. This approach not only preserves accurate target position information but, when combined with the small target detection head, allows the EMA module to leverage its capability of integrating channel and context information, thus addressing the occlusion problem of power line targets at different scales. The experimental results are shown in Table 4. As seen in Table 4, compared to the original YOLOv8s-seg model, the improved model with the added EMA attention module achieves a significant increase in accuracy. Specifically, the mAP@0.5 improves by 3.2%, thanks to the EMA module’s focus on reducing the computational burden on each channel. This results in a decrease in both the parameter count and computational load, with the number of parameters reduced by 0.64M, and FLOPs/G increased by 0.8G, thereby enhancing the model’s computational efficiency.

### 4.4. Ablation Experiment

To verify the enhancement effect of the improvement strategy proposed in this study on the target detection performance of power line images, this paper takes YOLOv8s-seg as the base model, introduces the above-mentioned improvement modules into its base network structure one by one, trains the power line dataset one by one, and finally compares the training results with the improved model Line-YOLO, and the experimental results are shown in Table 5.

The positions marked with a check indicate that the corresponding improvements are applied to the original YOLOv8s-seg model. As shown in the data from Table 5, the baseline model YOLOv8s-seg already demonstrates good performance in detecting the power line segmentation dataset. However, the improved model, Line-YOLO, outperforms the baseline model with a 6.2% increase in mAP@0.5. Additionally, the model’s parameter count is reduced by 2.05 M, the floating-point operations per second (FLOPS) are increased by 6.5 G, and the frames per second (FPS) for detection are improved by 14. Overall, the proposed improved model exceeds the baseline YOLOv8s-seg in terms of comprehensive performance metrics, achieving better detection accuracy. This demonstrates that the proposed method not only enhances detection precision but also achieves a lightweight model, fulfilling both real-time and accuracy requirements.

### 4.5. Comparison Experiment

To evaluate the performance of Line-YOLO, this study conducts a comparative analysis with existing popular object detection and lightweight models in a uniform experimental environment. From the Two-stage R-CNN to the One-stage YOLO series, the comparative data show that while the R-CNN model performs better in accuracy than most YOLO series models, it is significantly outperformed by YOLOv8s-seg. Among the YOLO series, YOLOv5s-seg and YOLOv7s-seg show a noticeable performance gap compared to YOLOv8s-seg. Therefore, this study ultimately selects YOLOv8s-seg as the baseline model for the experiment. The experimental results are shown in Table 6. Line-YOLO outperforms other network models in all accuracy metrics, particularly achieving an accuracy rate of 99.1% and an mAP@0.5 of 91.7%.

From the analysis of experimental data, it is evident that the improved Line-YOLO model achieves significant improvements in various performance metrics compared to other models when trained on the power line dataset. The precision curve (P curve), recall curve (R curve), F1 curve, and PR curve for the Line-YOLO model trained on the power line dataset are shown in Figure 6.

### 4.6. Convergence Analysis

The loss function measures the difference between the model’s predictions and the true labels. A decrease in the loss function indicates an improvement in the model’s robustness. The training result curve of the improved Line-YOLO model is shown in Figure 7. From the curve, it can be observed that as the number of iterations increases, the value of the loss function decreases rapidly and eventually stabilizes, indicating that the model has reached a convergent state.

From the training curve shown in the figure, it is clear that after approximately 50 iterations, the training curve tends to stabilize. The model’s loss remains low most of the time, and there is no occurrence of overfitting or underfitting. The loss value remains at a low level, and the accuracy is consistently maintained above 90%. This indicates that the model is well adapted to the training data and can generalize to unseen data to a certain extent.

### 4.7. Visualization Results

To intuitively demonstrate the performance of the improved algorithm in this study, the optimal weights obtained after training the YOLOv8s-seg model and the improved Line-YOLO model are used for testing. Three sets of images are selected for comparison. The results obtained after analyzing the same image data are shown in Figure 8. Specifically, (a) is the original image, (b) shows the detection result using the YOLOv8s-seg algorithm, and (c) displays the detection result using the Line-YOLO model. From the figure, it is evident that the YOLOv8s-seg model has lower detection accuracy for power lines and is prone to missing detections. In contrast, the Line-YOLO model provides a more comprehensive detection of power lines in the image, effectively detecting small target features and showing strong resistance to various interferences in the background environment of the distribution cabinet. This is particularly noticeable in the detection of small target power line segments on the surface of the distribution cabinet, making it more suitable for power line detection in such environments.

Based on the training of the Line-YOLO model on the power line dataset, the coordinates of the power lines in the distribution cabinet can be identified, and the optimal weights “best.pt” are obtained. The identification of power line positions is shown in Figure 9.

The detected power lines are divided into horizontal and vertical sections. The distribution box in the distribution cabinet serves as the horizontal reference line, equivalent to the x-axis in the coordinate system, as shown in Figure 10. If no target distribution box is present in the image, the shooting angle in the image is used as the horizontal direction.

The power line segmentation dataset is run in the Line-YOLO model for power line identification, resulting in the optimal weights “best.pt”. The custom-written “detect.py” program is then used to invoke the “best.pt” weights for detecting power line angles, with the detection principle described as follows:(1)Initially, color filtering is applied to the power lines. The color of the power lines is selected using the HSV color space, followed by morphological operations (erosion and dilation) to further extract the contours of the power lines. Next, the Hough transform is used to detect straight lines in the image. By calculating the angular difference between these lines and the horizontal axis of the image, the baseline angle of the power lines is determined.(2)Once the baseline angle is determined, the program continues to process the power line region, focusing on detecting the angle of horizontal power lines. Gaussian blur and adaptive thresholding are applied to the power line region to enhance the image contrast and remove noise. The Hough line transform is then used to identify horizontal lines, and their angles are calculated. By comparing the angle of the horizontal power line with the baseline power line, it is determined whether the horizontal power line is tilted. If the angular difference between the horizontal power line and the baseline is within 5°, it is classified as “Normal”. If the difference exceeds 5°, it is classified as “Possible skewness”.(3)Next, the angles of vertical power lines in the image are analyzed, categorized into left-upper, right-upper, and bottom vertical power lines. Gaussian blur and adaptive thresholding are again used to remove noise and enhance edges. The Hough transform is applied to detect vertical lines and calculate their angles. The angle of the vertical lines is compared with 90° (the ideal vertical angle) to determine whether the power lines are vertical. If the angular difference between the vertical power line and 90° is within 5°, it is classified as “Normal”. If the difference exceeds 5°, it is classified as “Possible skewness”.

The measurement results are illustrated in Figure 11. Among them, (a) and (b) are the images after detection, showing the recognition of the power lines and reference line, as well as the judgment result in the top-left corner; (c) is the software terminal screen, where information related to the detected images is displayed. The reference line angle, horizontal power line angle, and vertical power line angle are sequentially output, followed by the corresponding judgment.

## 5. Conclusions

In this paper, an improved algorithm based on YOLOv8s-seg: Line-YOLO is proposed, aiming at the task of power line angle detection. By adding deformable convolution, a BiFPN structure, an EMA module, and a small target detection head to YOLOv8s-seg, the performance and generalization ability of the model are improved. In this paper, relevant power line images are collected for the dataset, and data enhancement is performed on the dataset by applying commonly used image data enhancement methods to form a high-quality power line dataset. Then, these images are manually labeled to make sufficient preparations for the experiments.

In this paper, the structural features of YOLOv8 and its innovations are described in depth by comparing the earlier YOLO series of algorithms. To realize the accurate identification of power lines in distribution cabinets, corresponding improvements are also made based on the YOLOv8s-seg model to improve the detection performance of the network. First, the deformable convolutional DCNv4 is introduced into the backbone network of the YOLOv8s-seg network structure and fused with the C2f module to strengthen the performance of the model for small target detection. Then, optimization is applied to the Neck layer, and the BiConcat improvement module is obtained by introducing the BiFPN structure and fusing it with the Concat module, which reduces the time required for feature fusion. Then the EMA attention mechanism is introduced to increase the use of contextual information, which, by fusing with the small target detection head, not only improves the model’s ability to recognize the target but also effectively solves the problem of loss and misjudgment when the targets overlap.

In the experimental part, a single comparison experiment is first conducted using each improved module to demonstrate the effect of adding improved modules on the model performance. After that, ablation experiments and comparison experiments of other models are conducted to show the excellent performance of the improved model. Finally, the inspection results of the power line connection are shown, the vertical and horizontal angle measurements of the power line are made according to the inspection results, and the corresponding angle data are obtained. This indicates that Line-YOLO can fulfill the task of power line angle detection excellently, which provides support for the subsequent research to judge the connection quality of power lines.

## Figures and Tables

**Figure 1 sensors-25-00876-f001:**
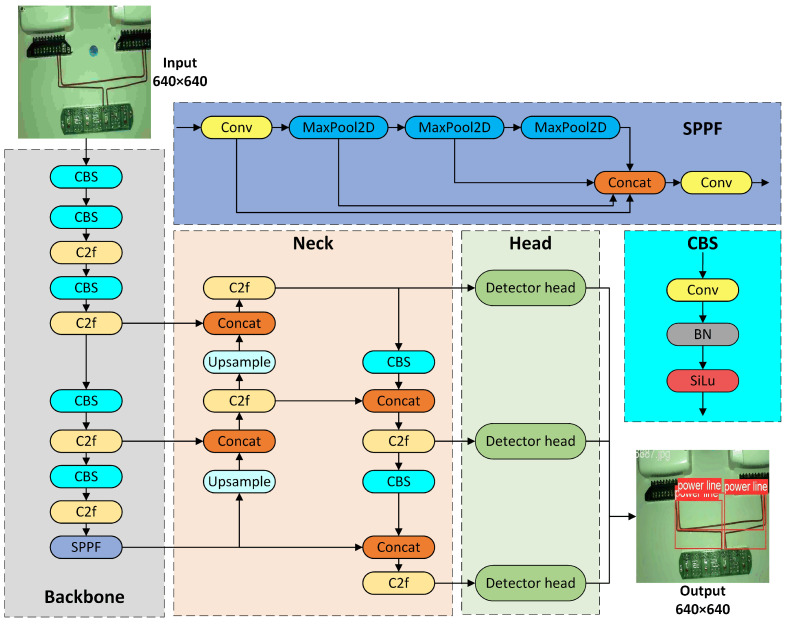
Line-YOLO network structure.

**Figure 2 sensors-25-00876-f002:**
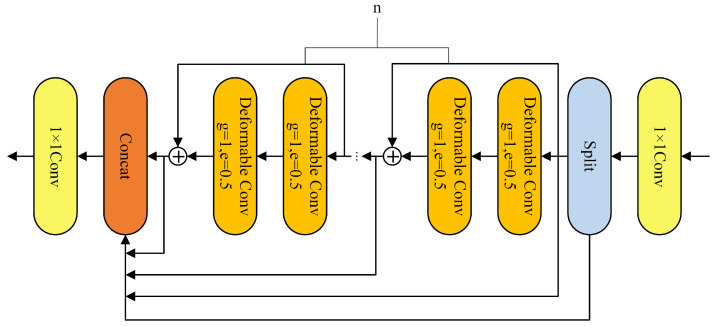
C2f_DCNv4 modular structure.

**Figure 3 sensors-25-00876-f003:**
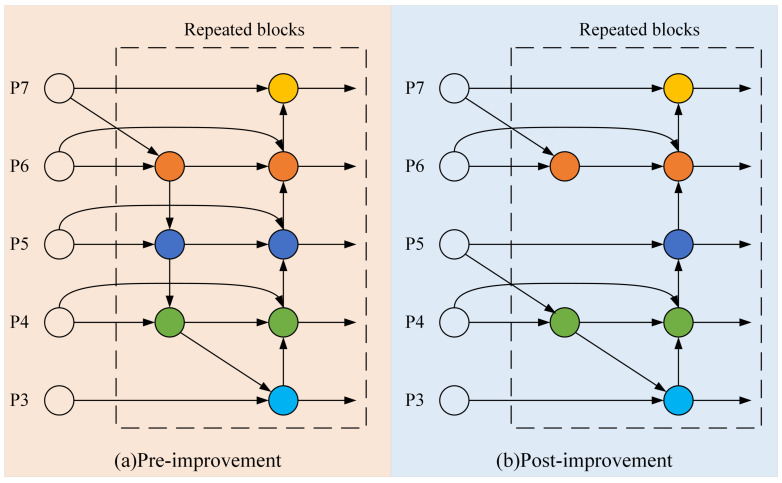
BiFPN structure.

**Figure 4 sensors-25-00876-f004:**
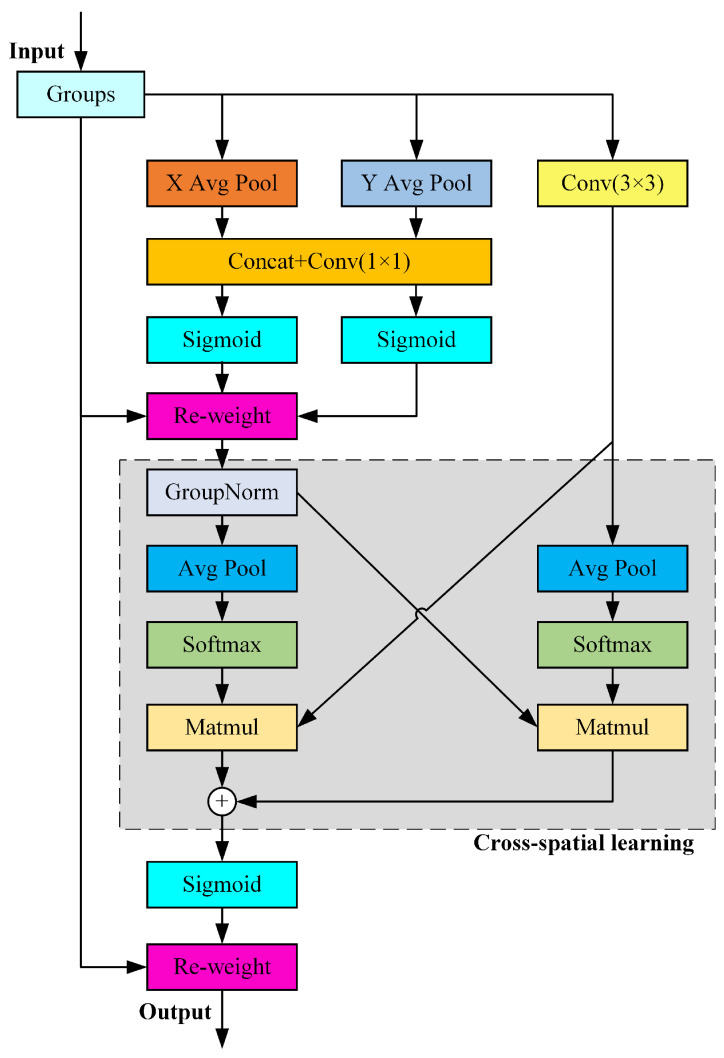
Structure of EMA attention mechanism.

**Figure 5 sensors-25-00876-f005:**
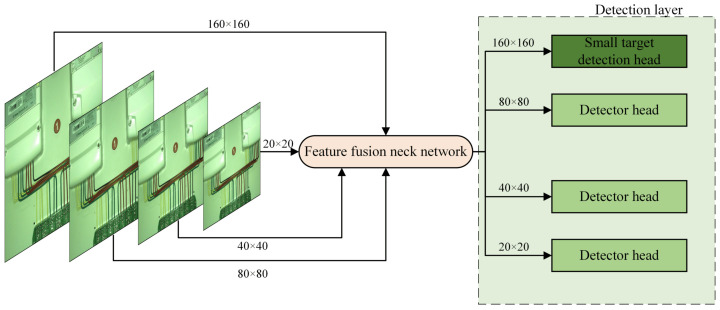
Improved detection layer.

**Figure 6 sensors-25-00876-f006:**
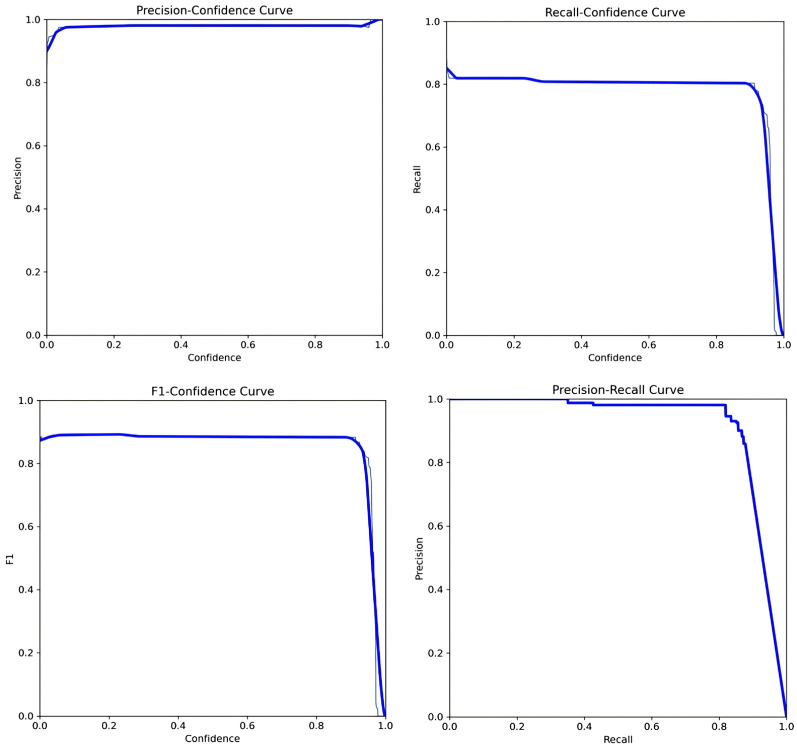
P Curve, R Curve, F1 Curve and PR Curve.

**Figure 7 sensors-25-00876-f007:**
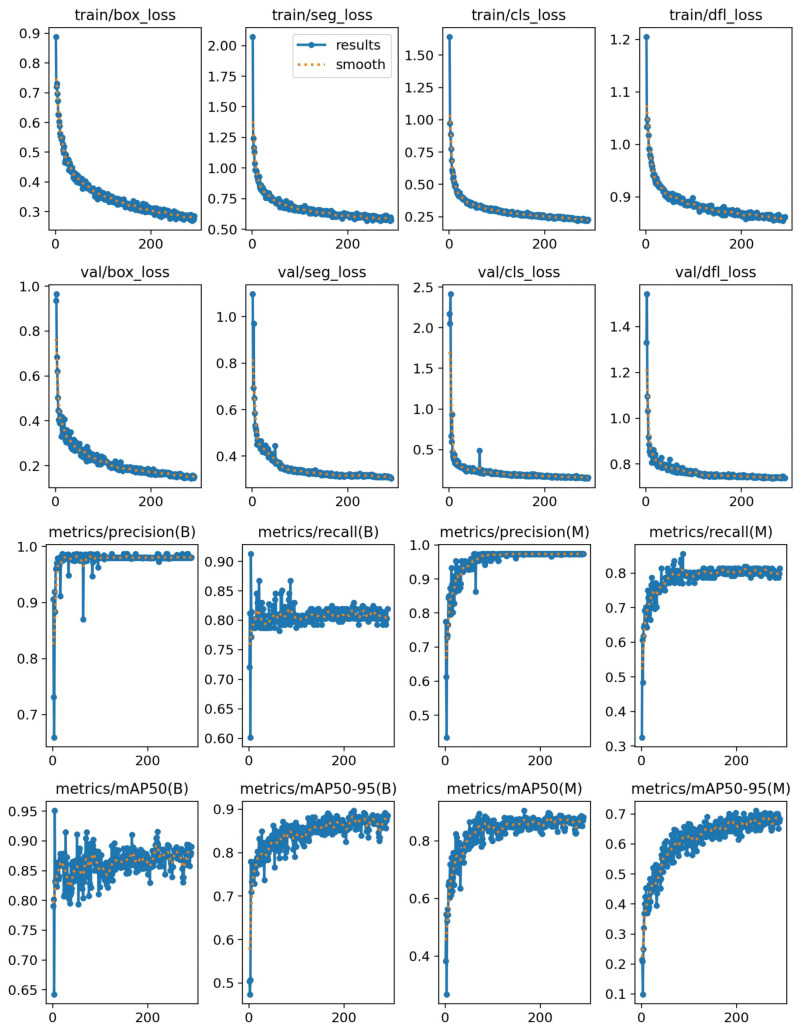
Running result of Line-YOLO model.

**Figure 8 sensors-25-00876-f008:**
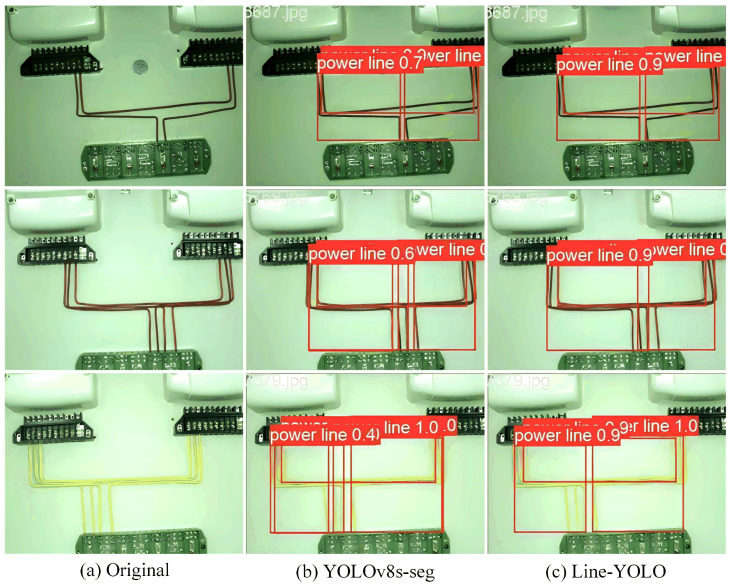
Different models test picture effect comparison.

**Figure 9 sensors-25-00876-f009:**
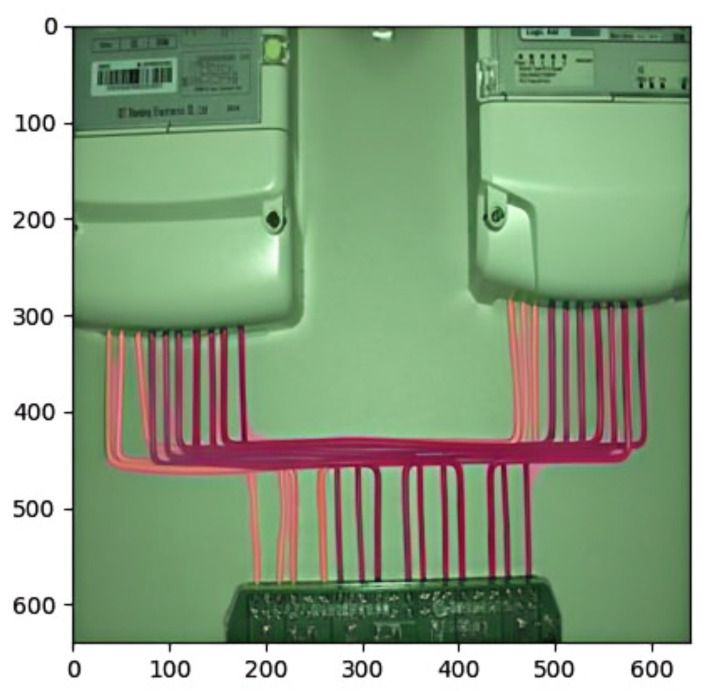
Identify power line locations.

**Figure 10 sensors-25-00876-f010:**
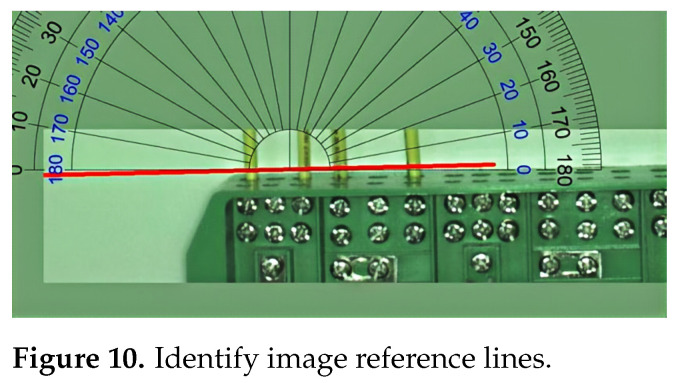
Identify image reference lines.

**Figure 11 sensors-25-00876-f011:**
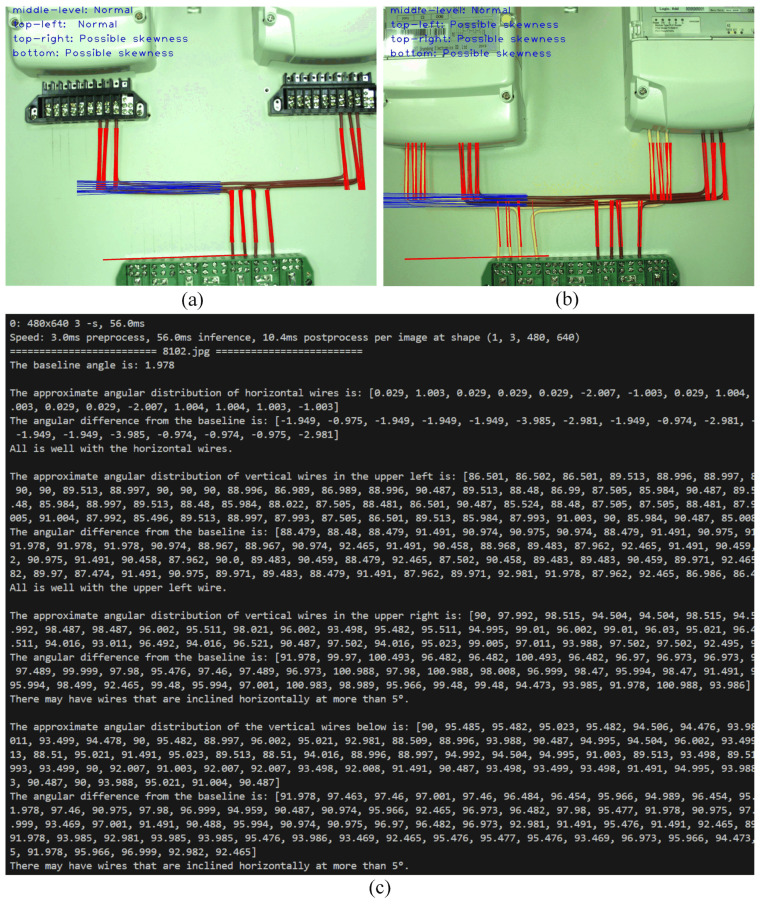
Power line angle measurement results.

**Table 1 sensors-25-00876-t001:** Hyperparameter settings.

Hyperparameters	Value
Input	640 × 640
Batch_size	16
Epoch	300
Learning_rate	0.01
Momentum	0.037

**Table 2 sensors-25-00876-t002:** Improved comparison of experimental results of C2f_DCNv4.

Model	mAP@0.5 (%)	Params (M)	FLOPs (G)	FPS
YOLOv8s-seg	85.5	7.26	18.4	77
v8s-seg+DCNv1	86.1	7.94	20.3	80
v8s-seg+DCNv2	86.8	8.31	21.1	81
v8s-seg+DCNv3	87.3	8.89	19.8	79
v8s-seg+DCNv4	87.6	9.34	19.3	81

**Table 3 sensors-25-00876-t003:** Comparison of different fusion strategies for BiFPN.

Model	mAP@0.5 (%)	Params (M)	FLOPs (G)	FPS
Weight	86.5	9.51	10.5	70
Sum	83.2	11.4	14.6	77
Adaptive	85.3	8.16	13.2	86
Concat	87.1	6.33	15.6	79

**Table 4 sensors-25-00876-t004:** Comparison of improvement of attention mechanisms with EMA.

Model	mAP@0.5 (%)	Params (M)	FLOPs (G)	FPS
YOLOv8s-seg	85.5	7.26	18.4	77
v8s-seg+SENet	85.7	4.58	18.3	78
v8s-seg+ECANet	87.1	7.42	18.8	83
v8s-seg+CBAM	85.4	7.18	18.0	80
v8s-seg+EMA	88.7	6.62	19.2	82

**Table 5 sensors-25-00876-t005:** Results of ablation experiment.

Model	C2f_DCN v4	BiConcat	EMA	Detector Head	mAP@0.5 (%)	Params (M)	FLOPs (G)	FPS
v8s-seg					85.5	7.26	18.4	77
A	✓				87.6	9.34	19.3	81
B		✓			87.1	6.33	15.6	79
C			✓	✓	88.7	6.62	19.2	82
D	✓	✓			86.2	5.69	14.7	79
E	✓		✓	✓	88.4	8.83	20.2	86
F		✓	✓	✓	88.1	5.72	18.3	81
Line-YOLO	✓	✓	✓	✓	91.7	5.21	24.9	91

**Table 6 sensors-25-00876-t006:** Different models compare experimental results.

Model	Precision (%)	Recall (%)	mAP@0.5 (%)	Params (M)	FLOPs (G)
R-CNN	94.2	85.6	88.4	40	33.5
Faster-RCNN	97.6	88.2	92.1	33.1	47.6
SSD	79.8	59.3	58.4	17.4	29.4
YOLOv5s-seg	80.6	77.8	83.1	6.83	7.9
YOLOv7s-seg	88.2	76.7	84.7	5.28	8.4
YOLOv8s-seg	93.4	83.6	85.5	7.26	18.4
YOLOv10s	93.2	81.3	85.2	7.19	21.6
YOLOv11s	94.7	83.2	86.4	9.41	21.5
YOLOv11s-seg	97.1	85.7	89.1	10.06	35.3
Line-YOLO	99.1	85.0	91.7	5.21	24.9

## Data Availability

The data presented in this study are available upon request from the corresponding author.

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
