# Peer review of "Line-YOLO: An Efficient Detection Algorithm for Power Line Angle"

_sensors, 2025, doi:10.3390/s25030876_

Round 1

Reviewer 1 Report

Comments and Suggestions for Authors

In this paper, an improved algorithm based on YOLOv8s-seg, capable of detecting power line angles, is introduced. However, the paper appears to have been written somewhat hastily, making it challenging to provide a fair evaluation.

1. Lack of clear definition for power line angle detection task

The paper fails to offer a precise definition of the power line angle detection task. There is an absence of a clear description outlining the power line angle and the associated detection objectives. A brief mention is made in Section 4.7 Visualization Results (lines 532-538), but the problem descriptions and analysis remain ambiguous. The description, along with the papers structural arrangement, makes the task of power line angle detection unclear to readers.

2. Absence of clear related works statement

In Section 2.1, Power Line Detection, detection methods for power lines are introduced, but methods for power line angle detection are notably absent. Power line angle detection should be a distinct problem from power line detection, and the correlation between the two is not clearly illustrated.

3.Insufficient clarity in Algorithm Development and Proposed Approach

In Section 2.2, “Algorithm Development”, only the development of YOLO-based methods is discussed. Although power line angle detection is proposed from an object detection perspective, there are various network structures related to object detection. The materials in Section 2.2 do not adequately support the proposed approach.

4.Evaluation concerns
The proposed improved model is based on YOLOv8s-seg. However, the paper lacks diversity in results comparison. Furthermore, the description of the power line angle measurement results is vague, lacking a thorough analysis of angle accuracy. It is difficult to assess whether the algorithm addresses the intended problem or has practical applicability in real-world scenarios.

Comments on the Quality of English Language

The English could be improved to more clearly express the research.

Reviewer 2 Report

Comments and Suggestions for Authors

1.Regarding the image Fig 12.Power line angle measurement results, is there a more sensible way of placing the image to ensure that the article image is easy to read?

2.Expression is a bit redundant, could try to be more concise

3.Is there a better way to represent Table 5? It is not straightforward enough to represent A,B,C,D in large paragraphs.

Comments on the Quality of English Language

Overall the quality of the English language is not a problem, but it could be more polished

Reviewer 3 Report

Comments and Suggestions for Authors

This paper proposes the Line-YOLO algorithm, which incorporates deformable convolution, BiFPN structure, EMA attention mechanism, and a small target detection head to improve the accuracy, efficiency, and robustness of power line tilt angle detection, achieving significant performance gains in detection accuracy, speed, and model efficiency. However, there are still several issues that need to be addressed.

1.     Grammar and typographical errors exist in the article, please proofread carefully.

2.     Please supplement the dimensions of input and output data or features in the structural diagram.

3.     From Table 5, it can be seen that embedding C2f_DCNv4 on the baseline results in an accuracy of 87.6%, while using BiConcat on this basis reduces the accuracy to 86.2. Please analyze the reasons for this phenomenon.

4.     Please provide the names of different comparison methods in Figure 9.

5.     Please give more details about the dataset used in this paper. For example, the resolution of the images, the division of the training set, the validation set, and the test set, etc.

6.     The methods in this paper are more like a combination of multiple methods and are not very innovative.

7.     The figures in your paper are a bit blurry. Please consider replacing them with clearer ones.

Comments on the Quality of English Language

The English could be improved to more clearly express the research.

Reviewer 4 Report

Comments and Suggestions for Authors

More experiments are advisable.

Round 2

Reviewer 3 Report

Comments and Suggestions for Authors

All the issues I am concerned about have been properly addressed.